Gene and genome-centric analyses of koala and wombat fecal microbiomes point to metabolic specialization for Eucalyptus digestion

Shiffman Miriam E. 1
Soo Rochelle M. 1
Dennis Paul G. 2
Morrison Mark 3
Tyson Gene W. 1
Hugenholtz Philip p.hugenholtz@uq.edu.au 1
1 Australian Centre for Ecogenomics, School of Chemistry and Molecular Biosciences, The University of Queensland , Brisbane , Australia
2 School of Agriculture and Food Sciences, The University of Queensland , Brisbane , Australia
3 The University of Queensland Diamantina Institute, Translational Research Institute, The University of Queensland , Brisbane , Australia
Prentis Peter
Electronic publication date: 2017 Nov 16
Publication date: 2017
Volume: 5
Electronic Location ID: e4075
Received 2017 Jul 31; Accepted 2017 Oct 31
Copyright: ©2017 Shiffman et al.
Copyright year: 2017
Copyright holder: Shiffman et al.
License: This is an open access article distributed under the terms of the Creative Commons Attribution License, which permits unrestricted use, distribution, reproduction and adaptation in any medium and for any purpose provided that it is properly attributed. For attribution, the original author(s), title, publication source (PeerJ) and either DOI or URL of the article must be cited.
License URL: https://creativecommons.org/licenses/by/4.0/

Keywords: Koala, Wombat, Microbiome, Marsupials, PSM, Metagenomics, Herbivory, Eucalyptus

Funding: ARC discovery project grants DP120103498 DP150104202 Fulbright scholarship University of Queensland New Staff Research Start-Up Grant This work was supported by the ARC discovery project grants DP120103498 and DP150104202. Miriam E. Shiffman was supported by a Fulbright scholarship and Paul G. Dennis was supported by a University of Queensland New Staff Research Start-Up Grant. The funders had no role in study design, data collection and analysis, decision to publish, or preparation of the manuscript.

==============================
The koala has evolved to become a specialist Eucalyptus herbivore since diverging from its closest relative, the wombat, a generalist herbivore. This niche adaptation involves, in part, changes in the gut microbiota. The goal of this study was to compare koala and wombat fecal microbiomes using metagenomics to identify potential differences attributable to dietary specialization. Several populations discriminated between the koala and wombat fecal communities, most notably S24-7 and Synergistaceae in the koala, and Christensenellaceae and RF39 in the wombat. As expected for herbivores, both communities contained the genes necessary for lignocellulose degradation and urea recycling partitioned and redundantly encoded across multiple populations. Secondary metabolism was overrepresented in the koala fecal samples, consistent with the need to process Eucalyptus secondary metabolites. The Synergistaceae population encodes multiple pathways potentially relevant to Eucalyptus compound metabolism, and is predicted to be a key player in detoxification of the koala’s diet. Notably, characterized microbial isolates from the koala gut appear to be minor constituents of this habitat, and the metagenomes provide the opportunity for genome-directed isolation of more representative populations. Metagenomic analysis of other obligate and facultative Eucalyptus folivores will reveal whether putatively detoxifying bacteria identified in the koala are shared across these marsupials.

Introduction

The gut microbiota of mammals have been implicated as key players in the radiation of this group into a multitude of dietary niches, including herbivory of most plant species (Ley et al., 2008). In concert, many plants have evolved mechanisms to defend against animal herbivory, including production of toxic secondary metabolites (Wink, 1988). In the co-evolutionary arms race between plants and the animals that eat them, the intestinal microbiota is a pivotal tool in the animal arsenal (Ley et al., 2008). The koala (Phascolarctos cinereus) is an arboreal folivore (leaf-eater) with a highly specialized dietary niche, subsisting entirely on the foliage of Eucalyptus trees. This material constitutes a challenging diet, lacking in calories and protein, high in lignified fiber, and enriched in plant secondary metabolites (PSMs). Eucalyptus PSMs include essential oils, long-chain ketones, cyanogenic glycosides, and polyphenolic compounds such as tannins, formylated phloroglucinols, and flavonoids, with demonstrated activity as animal toxins and antimicrobials (Cork, Hume & Dawson, 1983; Eberhard et al., 1975; Eschler et al., 2000). The importance of the gut microbiome to the koala’s ability to digest Eucalyptus is corroborated by the practice of pap feeding, a form of vertical microbiota transfer in which the newborn joey consumes a microbe-rich, feces-like substance secreted by its mother (Osawa, Blanshard & Ocallaghan, 1993). Further, multiple, taxonomically diverse bacterial isolates from the koala gastrointestinal tract have been shown to degrade tannin-protein complexes in vitro (Osawa, 1990; Osawa, 1992; Osawa et al., 1995). However, beyond 16S rRNA-based profiling of four koalas (Alfano et al., 2015; Barker et al., 2013), no published studies have used culture-independent methods to study the koala gut microbiome to our knowledge. Here we use the term microbiome in its originally defined sense as a microbial community occupying a defined theater of activity (Whipps, Lewis & Cooke, 1988).

We hypothesized that the koala gut microbiota include populations with specialized metabolism that contribute to the host’s ability to digest Eucalyptus. In order to identify microbial lineages and functional pathways specific to the koala digestive system, we sought to compare it to that of its closest living relative, the wombat. While both animals belong to the suborder Vombatiformes within the marsupial order Diprotodontia, the wombat is an herbivore that subsists primarily on grasses and does not consume Eucalyptus (Phillips & Pratt, 2008; Rishworth, Mcilroy & Tanton, 1995) (Fig. 1). We used shotgun metagenomics to study the fecal microbiomes of a southern hairy-nosed wombat (Lasiorhinus latifrons) and a koala from the same zoo and, to generalize our findings, also used 16S rRNA gene amplicon profiling to characterize microbial community membership of samples from multiple koalas and wombats across three zoos. To our knowledge, this is the first culture-independent report of the wombat microbiome.

Figure 1 Phylogeny, diet, and digestive strategy of marsupials from the order Diprotodontia.

Marsupials are color-coded by dietary preference and classified by digestive strategy. Eucalypt leaves are used to indicate the four Eucalyptus folivores, with the number of leaves corresponding to the extent to which each species relies on Eucalyptus as a primary food source. Molecular phylogeny is based on Meredith et al. (2008) and Meredith et al. (2010). The suborder Vombatiformes comprising koalas and wombats is highlighted. Images of marsupials reproduced from Kirsch, Lapointe & Springer (1997), with permission from CSIRO Publishing.

We identified microbial populations in koala feces that are distinct from those found in wombat feces, which, based on their genome sequences, likely play key roles in degrading Eucalyptus secondary metabolites. This study highlights insight gained from complementing gene- or pathway-centric “genome-agnostic” analysis of metagenomic data with analysis of discrete population genomes extracted from complex community data.

Materials & Methods

Data collection

Fecal samples were collected from eight koalas (Phascolarctos cinereus), four southern hairy-nosed wombats (Lasiorhinus latifrons), and three common wombats (Vombatus ursinus). Samples were collected from three zoos in Queensland, Australia (Lone Pine Koala Sanctuary, Cairns Tropical Zoo, and Wildlife Habitat Port Douglas), with at least two zoos represented per species to diminish zoo-specific effects on the microbiome. To enable differential coverage binning of population genomes, samples were collected on multiple dates from an individual koala named Zagget (documented previously by Soo et al., 2014) and an individual southern hairy-nosed wombat named Phil, both residing at Lone Pine Koala Sanctuary, Brisbane, Australia. Samples were collected under ethical permit ANRFA/SCMB/099/14, granted by the Animal Welfare Unit, the University of Queensland, Brisbane, Australia. Samples were stored in Eppendorf tubes at −80 °C prior to downstream processing.

Samples collected for this investigation were subject to mechanical and chemical lysis prior to automated genomic DNA extraction. In brief, ≈50 mg sample was added to tubes with 0.7 mm garnet beads (MO BIO, Carlsbad, CA, USA) and suspended in 750 µL Tissue Lysis Buffer (Promega, Madison, WI, USA). Following 10 min of bead-beating at maximum speed, samples were pelleted by centrifugation. Finally, 200 µL of supernatant was used as the input for DNA extraction on a Maxwell 16 Research Instrument with the Maxwell 16 Tissue DNA Purification Kit (Promega, Madison, WI, USA) according to the manufacturer’s instructions.

Genomic DNA from the three previously sequenced samples from Zagget the koala was extracted using the MP-BIO FASTSPIN Kit for soil (BP Biomedicals, Santa Ana, CA, USA) and bead-beating, as previously described (Soo et al., 2014).

16S rRNA gene amplicon community profiling

The V6 to V8 region of the 16S rRNA gene was targeted using the universal primers 926F (5′-AAACTYAAAKGAATTGRCGG-3′) and 1392R (5′-ACGGGCGGTGWGTRC-3′) (Engelbrektson et al., 2010) ligated to Illumina adapter sequences. The Illumina 16S library preparation protocol (#15044223 Rev. B) was followed. In brief, in the first round of amplification, products of ≈500 bp were PCR-amplified from genomic DNA template (1 ng/µl) using 2X Q5 HotStart HiFidelity MasterMix (New England Biolabs, Ipswich, MA, USA) under standard PCR conditions. Resulting amplicons were purified using Agencourt AMPure XP beads (Beckman Coulter, Brea, CA, USA) and indexed with unique 8 bp barcodes using the Illumina Nextera XT V2 Index Kit Set A-D (Illumina FC-131-1002; Illumina, San Diego, CA, USA) under standard PCR conditions. Equimolar indexed amplicons were pooled and sequenced on the Illumina MiSeq platform at the Australian Centre for Ecogenomics (ACE), UQ, using paired-end sequencing with V3 300 bp chemistry according to manufacturer’s protocol.

Raw data was demultiplexed and processed using a clustering-based approach. In brief, following quality control of forward reads (primer trimming, quality trimming with Trimmomatic (Bolger, Lohse & Usadel, 2014) and hard trimming to 250 bp), resulting sequences were processed using QIIME’s pick_open_reference_otus.py workflow with default parameters (97% similarity) (Kuczynski et al., 2011). Output Operational Taxonomic Unit (OTU) tables were filtered to exclude clusters with abundances below 0.05%, and taxonomies of representative OTU sequences were assigned based on BLAST against the Greengenes 16S reference database (release 13_5) (McDonald et al., 2012) clustered at 97% similarity. Statistical plots of OTU data were constructed using STAMP v2.0.8 (Parks et al., 2014).

Metagenomic shotgun sequencing

Paired-end shotgun sequencing was used to analyze multiple sample time-points from an individual koala (Zagget) and wombat (Phil) residing at Lone Pine Koala Sanctuary, Brisbane, Australia.

As previously described by Soo et al., two of the three koala time-points (Zag_T1, Zag_T3) were prepared for sequencing with the Nextera DNA Sample Prep kit (Illumina, San Diego, CA, USA). Samples were sequenced on the Illumina HiSeq2000 platform one quarter of a flow cell each, producing 150 bp paired-end reads. The remaining sample (Zag_T2) was prepared with the TruSeq DNA Sample Preparation Kit v2 (Illumina, San Diego, CA, USA) and sequenced on one lane of a flowcell on the Illumina HiSeq2000 platform, with 150 bp paired-end reads.

Five wombat time-points (PhilSHN_T1-T5) were prepared with the Nextera DNA Sample Prep kit (Illumina, San Diego, CA, USA). Samples were sequenced on 2/11 of a flow cell each on the Illumina HiSeq1000 platform, producing 100 bp paired-end reads.

Metagenomic sequence assembly

Raw shotgun reads were subject to stringent quality control using SeqPrep (https://github.com/jstjohn/SeqPrep) for adaptor trimming and Nesoni v0.108 (https://github.com/Victorian-Bioinformatics-Consortium/nesoni) for quality filtering (with a minimum quality threshold of Phred score 20 and minimum read length of 30 bp). For each host, high-quality reads from all samples were co-assembled de novo in CLC workbench v7 (CLC Bio, Taipei, Taiwan) with automated k-mer size selection. To determine coverage of assembled contigs, high-quality reads from each sample time-point were mapped to their respective co-assembly with BWA (Li & Durbin, 2009).

Metagenomic community profiling

CommunityM v0.1.2 (https://github.com/dparks1134/CommunityM) was used to determine community profiles of shotgun metagenomic datasets based on 16S rRNA sequences. Briefly, reads corresponding to 16S or 18S rRNA sequences were identified with profile hidden Markov models (HMMs) and mapped with BWA-mem (Li & Durbin, 2009) to the Greengenes 97% (release 13_5) (McDonald et al., 2012) and SILVA 98% (release 111) (Quast et al., 2013) reference databases. Output OTU tables were filtered to exclude lineages with relative abundances below 0.05%. Krona plots were used to compare mean phylum-level community composition (Ondov, Bergman & Phillippy, 2011). Additional heatmapping and bidirectional clustering were conducted using STAMP v2.0.8 (Parks et al., 2014), and lineages most distinctive to each host were identified on the basis of effect size, among groups that differed significantly between hosts (Welch’s t-test, p < 0.05).

Community-level functional profiling

Predicted coding sequences (CDSs) were identified in the koala and wombat co-assemblies using Prodigal v2.6.0 in metagenomic mode (using the -meta flag) (Hyatt et al., 2010). Each co-assembly was assessed for carbohydrate-active enzyme (CAZyme) functional potential using a database of profile HMMs (Yin et al., 2012) generated from CAZy families (Cantarel et al., 2009) with standard E-value and coverage thresholds. To determine community-wide prevalence, each CAZy annotation was weighted by average read coverage of its associated contig, normalized to overall coverage of all CDSs, at each sample time-point. Identified glycoside hydrolases (GHs) were assigned to steps in plant fiber degradation according to the categories designated by Allgaier et al. (2010). In order to calculate the percentage of lignocellulolytic GHs among all CDSs, only unique CDSs were included to account for genes with multiple GH domains. Relative abundance of CAZy categories was compared across sample time-points using STAMP v2.0.8 (Parks et al., 2014).

Weighted CAZy profiles were also compared to those of other microbial ecosystems associated with lignocellulose degradation: non-Vombatiformes (wallaby foregut) (Pope et al., 2010), non-marsupial (cow rumen) (Brulc et al., 2009), non-mammalian (termite hindgut) (He et al., 2013), and non-host-associated (switchgrass-adapted compost) (Allgaier et al., 2010). Here, a single relative weighted profile for each marsupial host was computed based on mean coverage of contigs containing relevant protein coding sequences, averaged across sample time-points.

Predicted CDSs for each assembly were also annotated using the KEGG (Kyoto Encyclopedia of Genes and Genomes) Automatic Annotation Server (Moriya et al., 2007) with the single-directional best hit method. To determine community-wide prevalence, each KEGG Orthology (KO) annotation was weighted by average read coverage of its associated contig, normalized to coverage of all CDSs, at each sample time-point. KO hits were assigned to all corresponding KEGG pathways according to the KEGG metabolism hierarchy (categories 1.1–1.11) (Kanehisa & Goto, 2000). Pathways were filtered to exclude those with fewer than three assigned KOs, and cumulative normalized abundance of KEGG pathways was compared across sample time-points using STAMP v2.0.8 (Parks et al., 2014). Differential pathways for each host were identified using Welch’s t-test with Benjamini–Hochberg false-discovery rate correction (q < 0.05).

Population genome binning and evaluation

Population genomes were extracted from each co-assembly using GroopM v0.2.10.18, which bins contigs based on differential coverage patterns across related shotgun datasets (Imelfort et al., 2014). Wombat population genomes were extracted using default parameters (based on core bin recruitment with minimum contig length 1,500 bp), and koala population genomes were extracted by adjusting minimum contig length for core bin recruitment to 1,500 bp, 2,000 bp, and 2,000 bp with subsequent recruitment of 1,500 bp contigs.

Population genomes were evaluated for quality (estimated completeness and contamination) using CheckM v0.9.4 (Parks et al., 2015). In brief, completeness and contamination were estimated based on the presence/absence and count of lineage-specific single-copy marker genes. Population genomes were considered for further analysis if estimated completeness was ≥ 50% and estimated contamination was ≤ 10%.

Bacterial population genomes were assigned taxonomic lineages by constructing a maximum likelihood tree using FastTree v2.1.7 (WAG + GAMMA model, other parameters set to default) (Price, Dehal & Arkin, 2010) with 26,849 RefSeq reference genomes (release 76) (O’Leary et al., 2016) based on a concatenated set of 120 conserved bacterial marker genes (Soo et al., 2017). Bootstrapping (100 times using non-parametric bootstrapping) was performed on the population genomes and a subset of 41 closely related RefSeq reference genomes. The inferred tree was imported into ARB v5.5 (Ludwig et al., 2004) for visualization and exported to Adobe Illustrator for figure production. In addition, an archaeal population genome was assigned to a taxonomic lineage as described above, but with 544 archaeal RefSeq reference genomes based on a concatenated set of 122 conserved archaeal marker genes (http://gtdb.ecogenomic.org/downloads) and bootstrapped.

Community-wide abundance of each population genome at each sample time-point was estimated based on average read coverage of its associated contigs, normalized to total contig coverage, using CommunityM v0.1.2 (https://github.com/dparks1134/CommunityM).

For each genome, predicted CDSs were identified using Prodigal v2.6.0 (Hyatt et al., 2010). To compare pairs of population genomes, aggregate amino acid identity (AAI) was determined using methods adapted from Konstantinidis & Tiedje (2005). In brief, for each pair, orthologs were identified based on reciprocal best-scoring BLAST hits among genome CDSs, with minimum thresholds of 30% amino acid identity and 70% coverage of the alignable region across the shorter CDS. Mean and standard deviation values for AAI were then computed across orthologs.

Population genome functional profiling

Population genomes were assessed for functional potential as per community-level functional profiling, above. In brief, CAZy domains within CDSs were identified based on CAZy family HMMs (Cantarel et al., 2009; Yin et al., 2012), and assigned steps of lignocellulose degradation as per Allgaier et al. (2010). Predicted CDSs were also annotated based on bidirectional best hit BLAST against the KO database clustered at 70% amino acid identity using CD-HIT (Li & Godzik, 2006) to increase efficiency. Annotated KOs were assigned to all relevant KEGG pathways based on the KEGG metabolism hierarchy (categories 1.1–1.11) (Kanehisa & Goto, 2000). Pathways were filtered by minimum size of three KOs per pathway. Metabolic profiles of the top 15 populations from each community (based on median estimated abundance) were compared using STAMP v2.0.8 (Parks et al., 2014) by constructing principal component analysis (PCA) plots based on the relative abundance of KEGG metabolic pathways in each genome. Relative pathway coverage of each KEGG pathway in each genome was also calculated based on the number of KOs normalized to total KOs in a given pathway present across population genomes from both datasets. Heatmaps of population genome CAZy annotations and relative KEGG pathway coverage were constructed using STAMP v2.0.8 (Parks et al., 2014).

Average genome size evaluation

Average genome size (AGS) was evaluated using MicrobeCensus v1.0.5 (Nayfach & Pollard, 2015) on high-quality forward and reverse reads, with a read length trim of 100 bp. For each community, AGS was estimated across multiple biological replicates (sample time-points) and technical replicates (subsamples of one million quality reads). Statistical comparison of means (Student’s t-test) was carried out in RStudio v0.98.1080 (Racine, 2012) and plots were constructed using the R package ggplot2 (Wickham, 2009).

Nucleotide sequence data deposition

Population genome sequences are available in NCBI Genbank under BioProject PRJNA357304.

Results

Metagenome-based community profiling of fecal microbiomes

We shotgun sequenced bulk genomic DNAs extracted from fecal samples collected at three time-points from Zagget the koala and five time-points from Phil the wombat, producing 90.7 Gb and 22.0 Gb of metagenomic paired-end data, respectively (Soo et al., 2014; Table S1). To initially assess the community composition of the two fecal microbiomes, we identified reads encoding 16S rRNA genes from each metagenomic dataset and mapped them to the Greengenes 97% reference database (McDonald et al., 2012) (Tables S2–S3). Collecting fecal samples from each animal over multiple time-points improved our ability to discriminate between stochastic changes in the microbiome and true host differences.

Hierarchical clustering of samples by weighted community composition revealed a host signature across all time-points (Fig. 2A). Both communities were dominated by bacteria, with a greater average relative abundance of methanogenic archaea (genus Methanocorpusculum) in the wombat (2.14%) than in the koala (0.11%). We saw clear differences between hosts even at low taxonomic resolution (phylum level, Figs. 2B–2C) although both communities were dominated by Firmicutes and Bacteroidetes, at a mean ratio of 2:1 in the koala (standard deviation, or SD = 0.4) and 3.6:1 in the wombat (SD = 0.6). Most conspicuously, Proteobacteria, Synergistetes, and Cyanobacteria were overrepresented in the koala fecal community, while Spirochaetes and Tenericutes were distinctive of the wombat microbiome. Viewed at finer taxonomic resolution, we identified core microbial genera as well as genera specific to each host (Fig. 2A). Bacteroides was the most abundant single genus consistently present in both communities, alongside unclassified members of the families Ruminococcaceae and Clostridiales. Among genera with statistically significant differences between hosts, the lineages most distinctive to each host biome were identified on the basis of effect size. In the koala, these microbes comprised unclassified members of the Bacteroidales family S24-7 (Ormerod et al., 2016) (10.4% absolute difference in means; Welch’s t-test, p = 0.028), the order YS2 (Soo et al., 2014) (8.88%, p = 0.026), the family Desulfovibrionaceae (1.45%, p = 3.36E–4) and the family Enterobacteriaceae (0.862%, p = 6.50E–3). Microbes distinctive of the wombat fecal microbiome included unclassified members of the family Christensenellaceae (11.6%, p = 3.09E–4), the order Clostridiales (5.16%, p = 2.56E–3), and the genus Ruminococcus (5.09%, p = 4.07E–3). However, ≈20% of 16S rRNA gene sequences extracted from the koala datasets failed to map to the Greengenes database, compared to <10% of sequences from the wombat datasets, suggesting increased representation of novel diversity among koala microbiota (Welch’s t-test, p = 1.31E–3). Mapping of extracted sequences (which comprised both 16S and 18S rRNA genes) to the SILVA 98% database (Quast et al., 2013) confirmed that the great majority of these unmapped sequences were not eukaryotic (Table S4).

Figure 2 Community structure of Vombatiformes fecal microbiomes, at (A) genus, (B) phylum level for koala metagenomes and (C) phylum level for wombat metagenomes.

16S rRNA gene sequences were identified in metagenomic data and classified against the Greengenes database (McDonald et al., 2012). Prefixes (Y-axis in A) indicate rank; o, order; f, family and g, genus. Samples (X-axis in A) were hierarchically clustered based on weighted Bray–Curtis distances. Taxa not able to be classifed to genus level are indicated as unc (unclassified). No significant matches to eukaryotic (18S rRNA gene) sequences were detected based on mapping of reads to the SILVA database. Binned population genomes meeting quality thresholds for completeness and contamination are depicted in (A) as blue and orange circles, corresponding to population genomes extracted from the koala and wombat metagenomes, respectively. Black outlines indicate that the population genome is among the top 15 most abundant from each host analyzed in this report.

Metagenome assembly

In order to examine community-level metabolism, high quality reads from all samples for each host were co-assembled to maximize accurate identification of coding sequences (Imelfort et al., 2014). The resulting assemblies comprised 686.7 Mbp of wombat sequences and 305.2 Mbp of koala sequences (including scaffolds), with an N25 of ≈22 kb and ≈24 kb, respectively. The majority of high-quality reads (>75%) from each time-point could be mapped to their corresponding co-assembly (Table S5), verifying that each assembly was representative of its input data.

Comparison of lignocellulose degradation potential

We used the Carbohydrate-Active Enzyme (CAZy) database (Cantarel et al., 2009) to examine the potential of each fecal community to degrade plant cell wall material, a major component of both marsupial host diets. Cellulases are microbial enzymes that hydrolyze the β-1,4 linkages in cellulose, while endohemicellulases and accessory hemicellulases attack the backbone and side chains of hemicellulose, respectively. In turn, oligosaccharide-degrading enzymes degrade the downstream products of both processes. The mean lignocellulolytic glycoside hydrolase (GH) profiles identified in the koala and wombat microbiomes were comparable to other biomes associated with lignocellulose degradation, including other marsupial (wallaby foregut) (Pope et al., 2010), other mammalian (cow rumen) (Brulc et al., 2009), non-mammalian (termite hindgut) (He et al., 2013), and environmental (switchgrass-adapted compost) (Allgaier et al., 2010) (Table 1). We next sought to directly compare differences in microbiome-wide capacity for plant cell wall degradation across the koala and wombat datasets. Here, we included an additional category of auxiliary activity CAZymes (Agger et al., 2014), which degrade polyphenolic compounds such as lignin, a major component of plant cell walls that is indigestible by endogenous host enzymes and can structurally impede degradation of other plant material (Jung & Allen, 1995). Auxiliary activity enzymes were enriched in the koala microbiome (Fig. 3), which may act on the lignin and phenolic compounds enriched in Eucalyptus foliage relative to grasses consumed by wombats (Hume, 1999). The most dramatic differences however, were overrepresentation in the koala microbiome of oligosaccharide-degrading enzymes and other glycoside hydrolases (not implicated in plant cell wall degradation) (Fig. 3).

Metabolic pathway-centric analysis

In order to examine broader differences in community-wide metabolic potential between koala and wombat microbiomes, we conducted a pathway-centric analysis of both metagenomic assemblies based on cumulative normalized coverage of pathways from the KEGG database (Kanehisa & Goto, 2000). Comparison of metabolic pathway profiles across multiple time-points from both host microbiomes identified 43 and 23 differential pathways significantly overrepresented in the koala and wombat, respectively (Figs. S1–S2), after correcting for multiple hypothesis testing with the Benjamini–Hochberg method to reduce false-discovery rate (Welch’s t-test, q < 0.05). A number of overrepresented KEGG pathways in the koala are consistent with the CAZy analysis showing increased prevalence of oligosaccharide-degrading enzymes in this host (Fig. 3), including starch and sucrose metabolism (which includes resistant starches) and butanoate metabolism (a byproduct of bacterial fermentation of dietary fiber) (Fig. S1).

Table 1 Weighted profiles of glycoside hydrolases across koala and wombat fecal microbiomes, versus other environments.

Weighted profiles were computed based on the koala and wombat metagenomes sequenced in this study, and compared to those of other microbial ecosystems associated with lignocellulose degradation: non-Vombatiformes (wallaby foregut) (Pope et al., 2010), nonmarsupial (cow rumen) (Brulc et al., 2009), non-mammalian (termite hindgut) (He et al., 2013), and non-host-associated (switchgrass-adapted compost) (Allgaier et al., 2010). Predicted open reading frames (ORFs) from each co-assembly were annotated with glycoside hydrolase (GH) families according to hidden Markov models specified by the Carbohydrate Active Enzyme (CAZy) database (Cantarel et al., 2009). Annotations associated with the degradation of lignocellulose were then assigned their relevant role in the degradation process according to categories specified by Allgaier et al. (2010). Relative weighted profiles were computed based on mean coverage of contigs containing relevant protein coding sequences, averaged across sample time-points.

CAZy family	Known activity	Koala	Wombat	Wallaby foregut	Cow rumen	Termite hindgut	Compost	
Cellulases	
GH5	cellulose	6.9	7.5	3.7	1	13.9	3.2	
GH6	endoglucanase	0.0	0.0	0.0	0.0	0.0	2.1	
GH7	endoglucanase	0.0	0.0	0.0	0.0	0.0	0.1	
GH9	endoglucanase	1.5	4.3	0.0	0.9	4.3	4.3	
GH44	endoglucanase	0.5	0.2	0.0	0.0	0.8	0.4	
GH45	endoglucanase	0.0	0.0	0.0	0.0	0.6	0.0	
GH48	endo-processive cellulases	0.0	0.3	0.0	0.0	0.0	0.5	
Subtotal (%)		9.0	12.4	3.7	1.9	19.6	10.6	
Endohemicellulases	
GH8	endo-xylanases	0.5	1.1	0.4	0.5	2.7	0.5	
GH10	endo-1,4-β-xylanase	1.4	3.6	4.1	1.0	9.9	8.9	
GH11	xylanase	0.0	2.9	0.0	0.1	1.9	1.4	
GH12	endoglucanase & xyloglucan hydrolysis	0.0	0.0	0.0	0.0	0.0	0.6	
GH26	β-mannanase & xylanase	3.0	2.7	1.9	0.8	2.0	1.5	
GH28	galacturonases	4.6	4.4	0.7	0.6	1.4	0.9	
GH53	endo-1,4-β-galactanase	2.2	0.9	3.3	2.7	2.2	0.2	
Subtotal (%)		11.7	15.5	10.4	5.7	20.1	14.0	
Accessory hemicellulases	
GH16	xyloglucanases & xyloglycosyltransferases	2.5	4.5	1.5	0.1	0.6	2.0	
GH17	1,3-β-glucosidases	0.0	0.1	0.0	0.0	0.0	0.1	
GH51	α-L-arabinofuranosidase	4.5	2.7	4.5	9.9	2.0	7.8	
GH54	α-L-arabinofuranosidase	0.0	0.1	0.0	0.2	0.0	0.0	
GH62	α-L-arabinofuranosidase	0.0	0.0	0.0	0.0	0.0	1.7	
GH67	α-glucuronidase	0.2	0.5	1.9	0.0	3.3	3.6	
GH74	endoglucanases & xyloglucanases	1.0	2.6	0.4	0.0	0.7	1.6	
GH78	α-L-rhamnosidase	5.4	4.3	9.3	5.1	0.8	8.1	
GH81	1,3-β-glucanase	0.0	0.0	0.0	0.0	0.0	0.3	
Subtotal (%)		13.5	14.8	17.6	15.3	7.4	25.2	
Oligosaccharide-degrading enzymes	
GH1	β-glucosidase & other β-linked dimers	1.8	0.8	22.7	1.8	2.5	9.2	
GH2	β-galactosidases & other β-linked dimers	19.7	16.8	8.9	28.5	13.6	8.6	
GH3	mainly β-glucosidases	18.7	13.1	26.8	26.6	15.5	12.2	
GH29	α-L-fucosidase	3.3	6.3	0.7	4.2	1.2	2.1	
GH35	β-galactosidase	1.2	1.3	1.1	1.9	0.6	0.6	
GH38	α-mannosidase	0.9	1.9	1.1	2.6	4.2	2.6	
GH39	β-xylosidase	0.9	1.1	0.4	0.3	1.5	1.0	
GH42	β-galactosidase	0.8	1.1	3.0	1.9	6.9	2.5	
GH43	arabinases & xylosidases	18.5	14.8	3.7	9.3	6.6	11.3	
GH52	β-xylosidase	0.0	0.0	0.0	0.0	0.3	0.0	
Subtotal (%)		65.8	57.3	68.4	77.1	52.9	50.1	
Total lignocellulolytic GHs	3,024	7,494	264	651	653	801	
Percent of all ORFs...	0.85%	0.62%	0.71%	0.78%	0.78%	0.72%	

As anticipated, many koala differential pathways may be involved with degradation of known Eucalyptus plant secondary metabolites; namely, phenolic compounds (e.g., phenylpropanoid biosynthesis; benzoate degradation; phenylalanine metabolism; tropane, piperidine, and pyridine alkaloid biosynthesis; chlorocyclohexane and chlorobenzene degradation; tyrosine metabolism), long-chain ketones (synthesis and degradation of ketone bodies), and terpenes and terpenoids, the major components of Eucalyptus oil (limonene and pinene degradation, geraniol degradation) (Fig. S1). Notably, the aforementioned “biosynthesis” pathways may actually be used for degradation, depending on stoichiometry. Many of these pathways are not highly ranked when sorted by effect size (i.e., absolute difference between mean proportions) since this value is necessarily lower in pathways with fewer associated KOs. Nonetheless, these results suggest substantial differences in metabolic functional potential across hosts commensurate with known differences in diet.

Differential wombat pathways with the greatest effect size included methane metabolism, reflecting enriched archaea within the wombat community, and numerous core prokaryotic metabolic pathways (e.g., pyrimidine and purine metabolism, peptidoglycan biosynthesis, amino sugar and nucleotide sugar metabolism, and glycolysis/gluconeogenesis) (Fig. S2). We hypothesized that relative enrichment of these pathways in the wombat microbiome could reflect an overall paucity of the more exotic secondary metabolic pathways found in the koala. Thus, we predicted that expanded metabolic capacity among koala-associated populations would be reflected by commensurate differences in genome size. Indeed, estimation of average genome size (AGS) using MicrobeCensus (Nayfach & Pollard, 2015) revealed significant differences across hosts (Student’s t-test, p = 0.00250), with AGS of the koala community estimated to be ≈500 kbp larger than that of the wombat (Fig. 4). This is equivalent to ≈500 extra genes per average genome based on a prokaryotic gene density on the order of 1 gene per kilobase (Lane & Martin, 2010).

Figure 3 Potential for lignocellulose degradation across koala and wombat fecal microbiomes.

Weighted profiles of CAZymes across koala and wombat metagenomes. Predicted coding sequences (CDSs) were annotated based on the CAZy database (Cantarel et al., 2009) according to their role in lignocellulose degradation (glycoside hydrolases associated with breakdown of cellulose or hemicellulose and enzymes with auxiliary activities associated with breakdown of lignin), or as non-lignocellulolytic glycoside hydrolases. Relative weighted profiles were computed based on mean coverage of contigs containing relevant CDSs and normalized to coverage of all CDSs. Differences across sample time-points from each marsupial host were then tested for significance using STAMP (Parks et al., 2014).

Figure 4 Metabolic differences between koala and wombat fecal microbiomes may reflect differences in average genome size.

MicrobeCensus (Nayfach & Pollard, 2015) was used to analyze average microbial genome size per community across biological replicates (sample time-points) and technical replicates (subsets of one million high-quality reads drawn from forward and reverse reads from each time-point). Estimated average genome size within the koala fecal community is 506.7 kilobases larger than the wombat fecal community (p = 0.00250, t-test).

Zeroing in on koala- and wombat-associated population genomes

In order to understand community function from a population standpoint, assembled contigs were binned into population genomes based on differential coverage across sample time-points (Imelfort et al., 2014). After filtering bins for quality, 38 and 15 population genomes with ≥50% completeness and ≤10% contamination were obtained from the wombat and koala metagenomes, respectively (Table S6). To make a more equivalent comparison, the top 15 populations from each community (based on highest estimated median abundance across time-points) were considered for further analysis. These genomes were taxonomically classified by constructing bacterial and archaeal phylogenetic trees with 26,849 RefSeq genomes and 544 RefSeq genomes, respectively (Fig. 5 and Table S6). The top population genomes are phylogenetically diverse and represent 31.2% and 26.8% of the total koala and wombat metagenomes, respectively, based on read mapping (Fig. 6), and 48% and 69% of the families detected by 16S analysis in the koala and wombat, respectively (Figs. 2A, 6, and Table S6).

Figure 5 Genome-based phylogeny of population genomes extracted from shotgun sequencing of the koala and wombat fecal microbiomes.

(A) Top bacterial population genomes extracted from the microbiomes of the koala (red text) and wombat (blue text) were initially classified against a reference set of 26,849 RefSeq genomes (O’Leary et al., 2016). A bootstrapped maximum likelihood tree (Price, Dehal & Arkin, 2010) was then constructed using 41 reference genomes most closely related to the population genomes based on a concatenated set of 120 conserved bacterial marker genes (Soo et al., 2017). (B) The single archaeal population genome recovered from the wombat community was similarly classified against a reference set of 544 archaeal RefSeq genomes, and a bootstrapped maximum likelihood tree constructed using 24 relatives and a concatenated set of 122 conserved archaeal marker genes. Population genomes are labeled as k (koala) or w (wombat) and their median abundance rank (1 to 15), followed by their most specific level of taxonomic assignment; o, order; f, family or g, genus. Black circles in the trees represent nodes with ≥ 90% bootstrap support, grey circles represent nodes with ≥ 70% bootstrap support, and white circles represent nodes with ≥ 50% bootstrap support.

Figure 6 Representation of (A) koala and (B) wombat fecal communities by population genomes.

(A) Family-level microbial community composition (of all families representing ≥ 1% for at least one time-point), based on median relative abundance of 16S rRNA reads extracted from metagenomic data. (B) Top quality population genomes corresponding to each family, sized and ordered by median relative abundance. The koala YS2 population genome (starred) has a median abundance of 0%, and is sized based on mean abundance. Family-level bar charts and population genome circles are colored according to their phylum-level affiliation.

Comparing metabolic potential among top population genomes

We next sought to examine whether core populations shared by both marsupial hosts provided core functionality, while differential populations were responsible for differential functionality. To this end, we analyzed the metabolic potential represented among members of each community by annotating population genomes with a comparable workflow to that described for KEGG annotation of whole metagenome assemblies. Population genomes were compared by considering each as a collection of metabolic pathways with varying relative abundances. Principal component analysis (PCA) of top population genomes from both communities demonstrated inter-community diversity in metabolic potential. PC1 (explaining 30.4% of total variance) partitioned nearly all koala fecal microbiome populations (13/15) from most wombat fecal microbiome populations (8/15), while PC2 (explaining 18.1% of variance) was largely driven by differences between the wombat methanogen population (the only archaeal genome analyzed) and bacterial populations from both hosts (Fig. 7). Interestingly, two pairs of populations hailing from the two different hosts have nearly overlapping metabolic potential and represent related taxonomic lineages, down to the order- (Bacteroidales) and genus- (Ruminococcus) levels (Fig. 7). Consistent with this metabolic overlap, these pairs of genomes share considerable average amino acid identity (AAI): 52.5% AAI (SD = 13.7%, calculated across 1,339 orthologs) for Bacteroidales populations k08 and w13, and 55.9% AAI (SD = 14.8%, across 874 orthologs) for Ruminococcus populations k11 and w15. The most metabolically divergent populations in the wombat community belong to the Archaea (order Methanomicrobiales) and Tenericutes (order Mycoplasmatales) (Fig. S3B), which are also differential wombat lineages based on community structure (Fig. 2). Similarly, the most metabolically divergent populations from the koala microbiome are members of the Synergistaceae and families within the Proteobacteria (Desulfovibrionaceae and Rhodocyclaceae), likewise reflecting groups overrepresented in the koala with respect to community structure (Fig. 2 and Fig. S3A).

Figure 7 Inter-community diversity in metabolic potential among top population genomes from the koala and wombat.

The top 15 population genomes from the koala and wombat were annotated based on KEGG orthology (KO) (Moriya et al., 2007) and annotations assigned to their corresponding KEGG pathways. Principal component analysis was used to differentiate among populations by treating each genome as a collection of metabolic pathways with variations in relative abundance; thus, genomes with more similar distributions of metabolic potential appear closer in space. Orange circles correspond to population genomes from the wombat, and blue circles correspond to population genomes from the koala. Selected populations are labeled as per Fig. 5.

Metabolic potential relevant to host diet amongst top population genomes

We next sought to analyze the capacity of each population to metabolize relevant dietary compounds, particularly lignocellulose (common to both host diets) and Eucalyptus PSMs (enriched in or unique to the koala diet). Oligosaccharide-degrading enzymes comprised the most common category of lignocellulolytic glycoside hydrolases across population genomes from both hosts. By contrast, only a few populations are prominently equipped with cellulases—mostly Ruminococcus populations (k07, k11, w11, w15) –and hemicellulases—mostly Ruminococcus and members of the order Bacteroidales (k01, k08, w13; Fig. 8A). This finding implies that both gut communities depend on a subset of populations to hydrolyze complex plant polysaccharides to simple sugars for use by the greater majority.

Figure 8 Metabolic potential across the top 15 population genomes from the koala and wombat fecal communities.

Metabolic modules are represented along the y-axis: (A) steps in lignocellulose degradation (according to the CAZy database (Cantarel et al., 2009)), (B) secondary metabolic pathways hypothesized to intersect with the degradation of Eucalyptus compounds such as polyphenolics and oils, (C) core metabolic pathways, (D) components of the urea recycling pathway. Shading indicates relative abundance of a given pathway or gene cluster in each population genome according to the legend at the lower right. Population genomes are ranked by their median community abundance in their respective host (E). Genomes are labeled as per Fig. 5, and additionally estimated genome size (in megabases), and completeness are given in parentheses. Asterisks indicate the four koala populations with the greatest differential metabolic potential relative to the wombat community (based on the number of unique KO annotations relevant to all differential koala pathways, weighted by estimated genome completeness).

Population genomes that contributed most to the specialized metabolic capacity of the koala microbiome were identified based on the collective unique KO counts assigned to all differential koala pathways, inversely weighted by genome completeness. The greatest contributors to differential metabolism in the koala were phylogenetically diverse, comprising two different classes of Proteobacteria (affiliated with the families Rhodocyclaceae and Desulfovibrionaceae) as well as the families Lachnospiraceae and Synergistaceae (Fig. 8E, starred populations). These specialized koala fecal microbiome populations were distinct from those identified as prominent cellulose and hemicellulose degraders, suggesting that hydrolysis of complex plant cell wall material and putative detoxification of PSMs is largely stratified within the koala community (Fig. 8).

Unlike the differential pathways in the koala fecal microbiome, which were linked to secondary metabolism and compartmentalized between populations (Fig. 8B), wombat fecal microbiome differential pathways were more evenly distributed across the dominant population genomes from both hosts (Fig. 8C). As expected, methane metabolism was most highly represented in Methanomicrobiales from the wombat, though all populations contained components of this pathway that are common to other metabolic pathways (Fig. 8C).

We also examined the capacity of population genomes for urea processing by quantifying KOs for urea transport (i.e., the five subunits of the Urt membrane complex and/or the monomeric Utp protein) (Beckers et al., 2004; Bossé, Gilmour & MacInnes, 2001) and urea hydrolysis (the three-subunit urease enzyme and additional accessory proteins) (Mobley, Island & Hausinger, 1995). In the koala, the most abundant population genome, a member of the family S24-7 representing ≈8% of the community, encoded the full suite of ureolysis genes, including the Utp protein, a bacterial homolog of the eukaryotic urea transporter previously identified in Proteobacteria (Sebbane et al., 2002). A second, less abundant S24-7 population in the koala lacked all genes for ureolysis, suggesting divergence in functional niche between these relatives, which share substantial AAI (58.2%, SD = 15.4%, across 1,376 orthologs), just below the cutoff of 60% AAI typical of organisms grouped at the genus level (Luo, Rodriguez-R & Konstantinidis, 2014). In addition, one of the koala Ruminococcus populations encoded the complete urease complex and accessories, though it lacked an identifiable transporter. Among the dominant wombat populations only Succinivibrio has the capacity for urea transport and degradation (Fig. 8D). The koala metagenome also yielded a Succinivibrio population genome that was only 44% complete (and therefore excluded from comprehensive analysis) but nonetheless encoded three subunits of the urea transporter. The remaining subunits of the transporter and urease complex were identified among unbinned contigs, although they could not be definitively linked to the Succinivibrio population genome. The distribution of urea processing genes in both animal hosts suggests partitioning of this function to specific community members, as seen for plant polymer hydrolysis and PSM detoxification.

16S rRNA gene amplicon profiling of multiple koalas and wombats

In order to expand our findings beyond the microbiomes of the two individual animals, we used 16S rRNA gene amplicon sequencing to profile the gut microbial communities of eight koalas (including Zagget), four southern hairy-nosed wombats (including Phil), and three common wombats (Vombatus ursinus) from three Australian zoos. Amplicon sequencing generated an average of 188,934 forward reads per sample (43,170 to 401,721), representing 310 operational taxonomic units (OTUs) with ≥0.05% relative abundance. The individual shotgun sequenced koala and wombat samples were broadly representative of their respective species. All wombat samples clustered together; however, the microbial profiles of Lasiorhinus and Vombatus were not significantly different from one another and also did not reflect zoo location of the animals. Fecal profiles of the eight koalas showed greater variation, which was not explained by zoo-specific effects (Fig. 9). Of the four populations in Zagget inferred to be important in PSM detoxification (asterisked in Fig. 8E), only Synergistaceae was both prevalent and abundant among the surveyed koalas (Fig. 9). Synergistaceae were absent in the majority of wombat samples, further implicating this lineage as providing specialized functionality to the koala gut microbiome. The dominant population genomes recovered from Zagget and Phil, belonging to families S24-7 and Christensenellaceae, respectively, were highly discriminatory between the marsupial cohorts. Indeed, S24-7 may be underrepresented in Zagget relative to other koalas (Fig. 9), and warrants further investigation into its role within the koala gut community. No single microbial OTU was ubiquitous to all koalas sampled (Fig. 9), suggesting that multiple species serve overlapping roles in detoxification of Eucalyptus PSMs in the broader koala population. This hypothesis is supported by the finding of multiple population genomes encoding putatively redundant secondary metabolic pathways within Zagget (Fig. 8).

Figure 9 Community structure of fecal microbiota from multiple captive koalas and wombats.

Fecal communities of eight individual koalas, four common wombats, and three southern hairy-nosed wombats were profiled using 16S rRNA gene amplicon sequencing. Five biological replicates from Phil the southern hairy-nosed wombat and three biological replicates from Zagget the koala were averaged to represent a single column per individual host. OTUs were aggregated at the family level and starred taxa comprise population genomes in the koala community with the greatest differential metabolic potential (see Fig. 8). Samples were hierarchically clustered based on weighted Bray–Curtis distance at the family-level. Abbreviations; o, order; f, family; unc, unclassified.

Discussion

Koalas are highly specialized folivores, feeding solely on Eucalyptus leaves, which contain many toxic plant secondary metabolites (PSMs) to discourage folivory. The koala has evolved a number of adaptations to deal with these toxic compounds, including endogenous enzymes produced in the liver (Ngo et al., 2000). The koala’s gut microbiota are also thought to play an important role in both detoxification and digestion; however, studies have mostly been limited to cultured bacteria, some of which have been shown to detoxify PSMs (Osawa, Blanshard & Ocallaghan, 1993; Osawa et al., 1995; Singh et al., 2015) but may not be representative of the broader gut community. In order to gain holistic insight into the structure and function of the gut microbiome of the koala (Phascolarctos cinereus), we used culture-independent metagenomics and community profiling to compare it to a close relative with a generalist herbivorous diet, the southern hairy-nosed wombat (Lasiorhinus latifrons) (Fig. 1). Multiple fecal samples were collected from a captive koala and wombat at the same zoo in Brisbane, Australia over the course of several months (six and two months, respectively). Fecal samples were previously shown by 16S rRNA amplicon profiling to be an adequate, non-invasive proxy for the gut microbiomes of hindgut fermenters, with some variation in relative community makeup but consistent membership across koala caecum, colon, and feces (Barker et al., 2013).

Plant cell wall material (cellulose, hemicellulose, and lignin) constitutes a large portion of the diets of both the wombat and the koala (Cork, Hume & Dawson, 1983; Rishworth, Mcilroy & Tanton, 1995). Because this material is largely resistant to mammalian enzymes, both marsupials rely on their associated hindgut microbiota to metabolize complex plant compounds into short-chain fatty acids that can be absorbed by the host (Barboza & Hume, 1992; Hume, 1999). Complementary gene- and genome-centric analyses underscored this shared metabolic capacity of both gut communities. Among mammalian microbiomes, the koala and wombat communities are notable for the highest relative proportion of cellulases: ≈9% and 12%, respectively, versus 4% in the wallaby foregut and 2% in the cow rumen (Table 1). This difference may be related in part to differences in the primary site of fermentation, i.e., an enlarged foregut (convergently evolved in the wallaby and cow) versus an enlarged hindgut (wombat and koala). Foregut fermentation is often modelled as a continuous stirred-tank reactor, whereas hindgut fermentation is more analogous to a plug-flow reactor with periodic movement of contents (Penry & Jumars, 1987). Consistent with this observation, the termite hindgut and compost communities, which are both periodic flow systems (Hume, 1999), also had high relative abundances of cellulases (≈20 and 11%, respectively, Table 1). Differences in plant substrate likely also contribute to observed differences in lignocellulolytic profiles. For example, the higher degree of hemicellulose in the wombat diet (Hume, 1999) may reflect the increased prevalence of xylanases (7.6% vs. 1.9% in the koala) (Table 1), and the higher proportion of lignin and phenolics in the koala diet is consistent with an increased prevalence of auxiliary enzymes including lignases, peroxidases, and tannases (Fig. 3).

Genome-centric analysis of the fecal datasets revealed that these functions were not evenly distributed across the microbial communities; rather, certain populations were specialized in lignocellulose digestion. The genus Bacteroides, comprising many known lignocellulose degraders (Ponpium, Ratanakhanokchai & Kyu, 2000), was found to be a prominent core microbial lineage, both abundant and prevalent across koala and wombat samples (Fig. 2A). Examination of the wombat Bacteroides population genome confirmed the presence of genes for lignocellulose degradation, including the greatest capacity for oligosaccharide metabolism across all genomes examined, though no related koala Bacteroides genome of sufficient quality was binned (Fig. 8). Further analysis of dominant population genomes in the koala and wombat revealed that both marsupials rely on multiple populations of Ruminococcus to hydrolyze cellulose to the disaccharide cellobiose for use by a multitude of diverse populations within each gut community. The Ruminococcaceae family, named for their prevalence in the cow rumen, are also known to play a cellulolytic role in many other mammals, including humans (Chassard et al., 2012), indicating long association of these specialist cellulolytic microbial lineages with metatherian and eutherian animals that consume plants. Other well known cellulolytic bacteria, such as members of the phylum Fibrobacteres prominent in the rumen and termite hindgut (Abdul Rahman et al., 2016), were not detectable in the surveyed marsupial feces.

Because urea recycling is known to play a role in the digestive strategy of both marsupial hosts (Penry & Jumars, 1987; Barboza, 1993), we also examined the capacity of population genomes for urea transport and degradation. Like many herbivores, the wombat recycles most of the urea pool produced in the liver to the gut for bacterial degradation to ammonia in order to sustain the wombat’s low nitrogen maintenance requirements. Folivores like the koala also rely on microbial ureolysis to buffer acids produced by conjugation of Eucalyptus PSMs in the liver (Hume, 1999). Urea processing ability was highly partitioned in the microbiomes, with only three of the 30 dominant populations capable of this functionality (Fig. 8). The identification of urease-containing Succinivibrio in both the koala and wombat is consistent with a previous report of Succinivibrionaceae bacterium WG-1 isolated from the Tammar wallaby with capacity for urea degradation (Pope et al., 2011), suggesting that this microbial lineage fills a conserved niche in urea recycling among diverse marsupials. Similarly, urease-containing S24-7 and Ruminococcus populations have been found in a diversity of animals, including humans, mice, guinea pigs, and cows (Ormerod et al., 2016; Wozny et al., 1977).

While plant cell wall hydrolysis and urea recycling are common denominators of both marsupial gut ecosystems, the koala diet is differentiated by an abundance of phenolics, terpenoids, and lipids (i.e., essential oils) (Hume, 1999). These categories comprise toxic compounds produced by Eucalyptus as a chemical defense against herbivory (Pass, Foley & Bowden, 1998). We observed enrichment of secondary metabolic pathways in the koala microbiome that may intersect with degradation of Eucalyptus metabolites such as phenolic compounds, long-chain ketones, and essential oils. This diversification in metabolic potential among koala microbiota is concomitant with significantly larger average genome size (Fig. 4), suggesting on average an increased genetic capacity to act on the greater chemical diversity in substrates encountered by koala microbiota. As with the other functions described, putative PSM-degrading pathways are partitioned by population. Specifically, we propose that key players in Eucalyptus compound metabolism among koala microbiota include members of the genus Desulfovibrio and the families Synergistaceae, Rhodocyclaceae, and Lachnospiraceae (asterisked populations in Fig. 8E).

Among these key bacteria, the most promising candidate as a core specialized member of the koala microbiota is the Synergistaceae population, both in terms of metabolic potential and prevalence in additional surveyed koalas (Fig. 9). The phylum Synergistetes, to which Synergistaceae belongs, is observed to be rare (<1%) in most gut and environmental ecosystems (Godon et al., 2005); however, the Synergistaceae population comprised >4% in the majority of koala microbiomes sampled, and was observed at >12% in one koala (Fig. 9). Synergistaceae has previously been observed as a prevalent member of the microbial community in a healthy koala but not in a diseased koala, and was found at even higher levels in the hindgut itself (17.0% in the caecum and 14.8% in the colon) than in the fecal pellet (6.1%) (Barker et al., 2013). Given the low protein content in Eucalyptus (Hume, 1999), it is unlikely that these enriched Synergistaceae populations serve primarily as protein degraders, as they do in many environments (Godon et al., 2005). The koala Synergistaceae population genome is differentiated by the highest pathway coverage for “benzoate degradation and synthesis” and “degradation of ketone bodies” among the 30 populations analyzed, as well as the most domains associated with CAZy “auxiliary activities” enzymes, which act on lignin and other polyphenolic compounds (Fig. 8B). Notably, the family Synergistaceae includes canonical examples of xenobiotic degradation by host-associated microbiota. Synergistes jonesii was isolated from a goat in Hawaii and shown to degrade mimosine, a toxic amino acid from the Hawaiian Leucaena plant (Jones & Megarrity, 1986; Allison et al., 1992). Another member of the Synergistaceae, strain MFA1 related to the genus Cloacibacillus and resident in animal gut environments, has been shown to degrade the plant toxin fluoroacetate through a previously unknown anaerobic dehalogenation pathway (Davis et al., 2012). Our findings suggest that members of the Synergistaceae may be able to act on a wider range of xenobiotics than previously appreciated, using different pathways to degrade PSMs present in Eucalyptus, which largely do not resemble amino acids or fluoroacetate.

Notably, none of the previously characterized koala gut isolates feature prominently in our analysis, including isolates identified as degraders of tannin-protein complexes (members of the families Streptococcaceae, Enterobacteriaceae, and Pasteurellaceae) (Osawa, Blanshard & Ocallaghan, 1993; Osawa, 1990; Osawa, 1992; Osawa et al., 1995) and cellulose (members of the families Bacillaceae and Pseudomonadaceae) (Singh et al., 2015). These lineages were neither common nor abundant among sampled koalas (Fig. 9) nor represented amongst the top population genomes in Zagget (Fig. 8), suggesting that they are minor players in detoxification by koala gut microbiota despite possessing relevant metabolic capabilities. It would appear, therefore, that current koala gut isolates are not representative of the true in situ community. Indeed, our results are consistent with previously published reports on koala fecal microbiomes based on 16S sequencing (Alfano et al., 2015; Barker et al., 2013). Beyond providing a more representative profile of the koala gut microbiota, our culture-independent approach includes the assembly of population genomes that may directly facilitate genome-directed culturing of strains of interest, as previously demonstrated by isolation of a key acetogen, Succinivibrionaceae bacterium WG-1, from the Tammar wallaby (Pope et al., 2011).

Conclusions

To our knowledge, this is the first metagenome-based analysis of both the koala and wombat, and will serve to inform broader investigations into marsupial gut microbiomes. By comparing the fecal microbiomes of the koala and wombat we have identified similarities that reflect herbivorous diets and differences indicative of dietary specialization in the koala. Our findings largely support previous studies that microbial phylogeny recapitulates metabolic potential (Snel, Bork & Huynen, 1999; Segata & Huttenhower, 2011). We find evidence of core lineages shared by both marsupials that serve core roles in lignocellulose degradation and urea recycling. In addition, we observe populations from the same order (Bacteroidales) or family (Ruminococcaceae) with mirroring metabolic pathway composition despite differing host ecology. Further, differential populations with respect to community structure are largely responsible for differential community-level functionality in the koala, as reflected by genomic potential. Specifically, we predict that a Synergistaceae population plays a key role in the koala’s ability to subsist on Eucalyptus foliage.

These predictions could be tested through metatranscriptomics of koala gut microbiota, either in a community context or as isolates, to confirm that pathways predicted to intersect with Eucalyptus PSM degradation are expressed in vivo, and upregulated in vitro under exposure to purified Eucalyptus compounds. Feeding trials using gnotobiotic mice (Turnbaugh et al., 2009) or woodrats (Neotoma lepida) (Kohl & Dearing, 2016; Magnanou, Malenke & Dearing, 2009), which more closely mimic the koala’s endogenous enzymes for detoxification (Hume, 1999), would facilitate these experiments. For example, gnotobiotic model hosts could be inoculated with koala fecal samples or reconstructed communities with predicted key members to determine whether microbiota alone would be sufficient to transfer the Eucalyptus tolerance phenotype. Better characterization of the microbial species and genes responsible for Eucalyptus degradation in the koala could be used to restore the gut microbiota of malnourished koalas raised in captivity, and lead to novel enzyme discovery for applications in biosynthesis or bioremediation. Studying koala microbiota could also lead to the development of probiotics for cattle to be administered alongside Eucalyptus toxins targeting methanogens (Cieslak et al., 2013) in order to reduce methane emissions from livestock, a major contributor to global warming.

Another promising avenue of investigation will be to expand this study to include the three other marsupial Eucalyptus folivores (greater glider, common ringtail possum and brushtail possum), which share the unique ability among mammals to digest Eucalyptus yet are not monophyletic (Fig. 1). It remains to be seen whether specialized microbiota identified in the koala are shared by all four Eucalyptus folivores, or if microbial lineages have independently co-evolved with each host (Brooks et al., 2016).

Supplemental Information

Figure S1 Metabolic pathways enriched in the koala metagenomic dataset, compared to that of the wombat

Predicted CDSs from each marsupial co-assembly were annotated based on KEGG Orthology (KO) using the KEGG Automatic Annotation Server (Moriya et al., 2007). KO annotations were weighted by average coverage of associated contigs per sample time-point and normalized to overall coverage of all CDSs. Annotations were assigned to all corresponding KEGG pathways according to the KEGG metabolism hierarchy (1.1–1.11), and pathways were filtered to exclude those containing fewer than three KOs. Relative cumulative prevalence of KEGG metabolic pathways from each marsupial host were then compared using STAMP (Parks et al., 2014). Differential pathways were identified based on significant difference between datasets (q < 0.05, Benjamini-Hochberg FDR correction) and ranked by effect size (difference in mean proportion).

Click here for additional data file.

Figure S2 Metabolic pathways enriched in the wombat metagenomic dataset, compared to that of the koala

Predicted CDSs from each marsupial co-assembly were annotated based on KEGG Orthology (KO) using the KEGG Automatic Annotation Server. KO annotations were weighted by average coverage of associated contigs per sample time-point and normalized to overall coverage of all CDSs. Annotations were assigned to all corresponding KEGG pathways according to the KEGG metabolism hierarchy (1.1–1.11), and pathways were filtered to exclude those containing fewer than three KOs. Relative cumulative prevalence of KEGG metabolic pathways from each marsupial host were then compared using STAMP (Parks et al., 2014). Differential pathways were identified based on significant difference between datasets (q < 0.05, Benjamini-Hochberg FDR correction) and ranked by effect size (difference in mean proportion).

Click here for additional data file.

Figure S3 Intra-community diversity in metabolic potential among top population genomes within each community, colored by phylum

The 15 most abundant quality population genomes from the (A) koala and (B) wombat communities were annotated based on KEGG Orthology (KO) (Moriya et al., 2007). Annotations were assigned to all corresponding KEGG pathways according to the KEGG metabolism hierarchy (1.1–1.11), and pathways were filtered to exclude those containing fewer than three KOs. Principal component analysis was used to differentiate among populations by treating each genome as a collection of metabolic pathways with variations in relative abundance: thus, genomes with more similar distributions of metabolic potential appear closer in space. Circles corresponding to plotted genomes are colored by phylum.

Click here for additional data file.

Table S1 Shotgun sequencing statistics of Vombatiformes fecal communities

Fecal samples representing multiple time-points from a captive koala and captive wombat were sequenced, generating a total of 90.7 Gb and 22.0 Gb of raw data, respectively.

Click here for additional data file.

Table S2 Taxonomic makeup (%) of Zagget the koala fecal microbiomes across sample time-points

Shotgun reads corresponding to 16S rRNA sequences were identified with HMMs and mapped to the Greengenes 97% database (McDonald et al., 2012), and the resulting community makeup was assessed at each taxonomic level (including unmapped reads). Lineages with less than 0.05% of corresponding reads were grouped and reported as “Other” (if the resulting group represented >0.05% of the community). Here, k, represents kingdom; p, represents phylum; c, represents class; o, represents order; f, represents family; and g, represents genus.

Click here for additional data file.

Table S3 Taxonomic makeup (%) of Phil the wombat fecal microbiomes across sample time-points

Shotgun reads corresponding to 16S rRNA sequences were identified with HMMs and mapped to the Greengenes 97% database (McDonald et al., 2012), and the resulting community makeup was assessed at each taxonomic level (including unmapped reads). Lineages with less than 0.05% of corresponding reads were grouped and reported as “Other” (if the resulting group represented >0.05% of the community). Abbreviations; k, kingdom; p, phylum; c, class; o, order; f, family; g, genus.

Click here for additional data file.

Table S4 The majority of 16S and 18S rRNA reads extracted from koala and wombat metagenomic samples are bacterial

Reads mapping to 16S and 18S rRNA genes were extracted from quality-filtered shotgun sequencing samples using hidden Markov models and mapped to the SILVA 98% database (Quast et al., 2013). Table values are kingdom-level percentages of all extracted reads. Here, lca is last common ancestor.

Click here for additional data file.

Table S5 Co-assembly statistics of Vombatiformes fecal communities

All time-points from each host dataset were co-assembled de novo (CLC) following stringent quality control of raw reads (adaptor trimming with SeqPrep and quality trimming with Nesoni). Co-assemblies were then validated by mapping high-quality paired reads from each time-point back to the corresponding assembly (BWA).

Click here for additional data file.

Table S6 Bin statistics of quality population genomes extracted from koala and wombat metagenomes, ranked by median abundance

Click here for additional data file.

We thank Varun Mazumdar, Donovan Parks, and Steve Robbins for providing valuable feedback on the study, and Jacqui Brumm at Lone Pine Koala Sanctuary, Cairns Tropical Zoo, Clare Anderson at Wildlife Habitat Port Douglas, and Emily Hoedt for facilitating collection of koala and wombat fecal samples. We also thank Nicola Angel and Serene Lowe for Illumina shotgun and 16S amplicon sequencing.

Additional Information and Declarations

Competing Interests

Author Contributions

Animal Ethics

Data Availability

The authors declare there are no competing interests.

Miriam E. Shiffman conceived and designed the experiments, performed the experiments, analyzed the data, contributed reagents/materials/analysis tools, wrote the paper, prepared figures and/or tables, reviewed drafts of the paper.

Rochelle M. Soo performed the experiments, analyzed the data, wrote the paper, prepared figures and/or tables, reviewed drafts of the paper.

Paul G. Dennis and Gene W. Tyson contributed reagents/materials/analysis tools, reviewed drafts of the paper.

Mark Morrison reviewed drafts of the paper.

Philip Hugenholtz conceived and designed the experiments, contributed reagents/materials/analysis tools, wrote the paper, reviewed drafts of the paper.

The following information was supplied relating to ethical approvals (i.e., approving body and any reference numbers):

Samples were collected under ethical permit ANRFA/SCMB/099/14, granted by the Animal Welfare Unit, the University of Queensland, Brisbane, Australia.

The following information was supplied regarding data availability:

The population genomes in this article have been uploaded to the NCBI BioProject accession number PRJNA357304.

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
