# Peer review of "Gene and genome-centric analyses of koala and wombat fecal microbiomes point to metabolic specialization for Eucalyptus digestion"

_PeerJ, doi:10.7717/peerj.4075_

## Round 0.1 · original submission · Minor Revisions

Overall I feel this is an excellent manuscript which provides much needed data about the microbiome of the koala and wombat. The project was well carried out and the statistical analysis has been conducted in thorough and rigorous way. The main point from the from the two reviews is that one feels the manuscript could be streamlined through the removal of redundant information in some sections of the manuscript (see review). Please also check the reference list, I found some small mistakes on line 966, 896, 726 and 706. Maybe if possible it would be nice to have a consistent size in the koala and wombat insets in figures. I liked the size in figure 3a, but this is completely optional. Please take care to respond to all of the reviewers comments and thank you for submitting such a high quality manuscript for review.

Reviewer 1 ·

Basic reporting

No comment

Experimental design

The majority of this project is excellent. This is the first in-depth culture-independent metagenome-based analysis of koala and wombat. A weakness relates to the use of fecal pellets as the sole source of microbiome analysis and then relating the microbiota to koala and wombat digestion. Eventhough there was an early publication in the koala, this only had a very low number of animals. The digestive microbiota communities in the gut and caecum and likely to be different to those in the fecal pellet. Can the authors perhaps include the comparisons of the metagenomes of a single animal.

Validity of the findings

No comment

Reviewer 2 ·

Basic reporting

The basic reporting in this paper is fine. Overall it is clearly written. My most substantial comment is that sometimes information appears in the wrong section and is sometimes redundant with what has already been stated. For example: Lines 80-90 present a summary of the results. This does not seem appropriate for the introduction. I have flagged several of these instances but the authors are encouraged to conduct a thorough review.

The figures for the most part are highly useful. Some required additional explanation. E.g.,
Fig 2: indicate the significance of the different number of circles in panel A.

Experimental design

The design and analysis of the results are well defined. The knowledge gap being addressed is articulated. The only criticism of the methods is that it would have been better if there were more control over the origin of the samples (ie more samples from the same zoo), but given the difficulty in keeping these species, the design is reasonable, and did not appear to impact the outcome.

Validity of the findings

Overall the findings seem valid, robust and statistically sound. The notion that there are members of the koala microbiome that function to degrade PSMs is likely true, and the authors for the most part acknowledge that this is a hypothesis yet to be experimentally tested.

Additional comments

In this manuscript, the authors present a comparative study on the gut metagenomes of two iconic mammals from Australia, the koala and the wombat. This is the first investigation into these metagenomes. They found that while there are many similarities in these metagenomes, the koalas is larger with greater representation of metabolism of plant secondary metabolites. The authors propose some key players in the microbiomes of these animals related to PSM metabolism, lignocellulose degradation and urea recycling.

S24-7 is mentioned in the abstract as being one of the unique members of the koala microbial community. The role of family is not further discussed in the discussion, yet it has been implicated in the degradation of PSMs, particularly oxalate, which likely occurs in Eucalyptus. Some references of potential interest: Ormerod et al 2016 4:36 Microbiome and Millar et al AEM 82:2669;

Along these lines, the putative function of the two taxa common in the wombat were not mentioned in the discussion.

The discussion could be improved through the inclusion of additional recent literature. recent experimental Such as Kohl et al Ecology Letters 2014; Welte, 2016 AEM 04054; Haiser et al Science 2013 341:295, Spanogianopoulos Nature Reviews Microbiology 2016 14:273

Figure 1 is beautiful. Does it include all genera in the Diprotodonts or is it an abbreviated tree?

Suggest removing “toxic” from Eucalypt. The concept of high levels of plant secondary metabolites can be explained in the text.

“’putatively” detoxifying bacteria

Lines 80-90 present a summary of the results. This does not seem appropriate for the introduction

Lines 275-283 mostly contain information that should be in the methods.

Fig 2: indicate the significance of the different number of circles in panel A.

The term CDS seems to be defined long after its first use

Fig 3: add an explanation of the use of colored dots. I figured it out eventually but it would be useful as an aid to the reader.

Lines 348-359: I could not identify the results in this paragraph. Seems like it belongs in the methods.

Interpretation that may be better suited to the discussion appears in the results, check lines 375-400 for examples.

The paper would be significantly shorter if the methods were not reproduced in the results

425: I think it would be more accurate to say microbe populations from koalas instead of koala populations; same comment line 471

443-455: most of this is methods

461-463: discussion

504-507 methods
524-527: discussion

619: restate the phylum’s name of Synergistaceae here

694-5: citation for phylosymbiosis? See Bordenstein’s work

---

## Round 0.2 · accepted · Accept

The authors have done an excellent job of addressing the reviewers comments and I feel that the manuscript is now acceptable for publication.